# Classification driven Assisted Screening for cervical cancer using deep neural network

**Harinarayanan K K**
Aindra Systems
Bangalore, India 560078
hari@aindra.in

**Nirmal Jith OU**
Aindra Systems
Bangalore, India 560078
nirmal@aindra.in

## Abstract

Automating cervical cancer screening has the potential to reduce high mortality rates due to cervical cancer, especially in developing countries. The most promising of these techniques is assisted screening in which preliminary analysis is validated by a pathologist. Assisted screening requires classification algorithms for initial screening that is later validated by a pathologist. It also needs attention, detection or segmentation algorithms for drawing attention of pathologists to important regions. Existing algorithms for cervical cancer screening focus on classification of individual cells . This focus leads to need for accurate segmentation of cells and inability to use extracellular information. In this work we propose a segmentation free deep learning algorithm for classification of PAP smear images. The proposed algorithm uses the intrinsic information in the network to generate a map of important regions for the pathologist to look into. This map is generated with sub image resolution while the training data contains annotations at the image level only. Our analysis on a dataset of around 14000 images validate the approach of assisted screening in reducing the pathologist workload by a large factor.

## 1   Introduction

Cervical cancer is one of the most prevalent cancer among women with more than half a million cases reported every year[1]. It has been observed that the use of systematic screening using Papanicolaou test (PAP-test) can reduce morality rate by as much as 70% or more[2]. However the current procedure for cervical cancer screening with PAP-test depends on high skilled labour. It is also a fatigue inducing process thereby limiting the number of slides that can be screened to a maximum of 100 per day according to clinical laboratory improvement amendments 1998 [3]. In light of these challenges, automation of cervical cancer screening is an important problem to consider. One of the most promising approaches for automation is assisted screening where algorithms provide initial results and assist a pathologist in validating/reviewing the result.

In this work, we propose a new deep learning algorithm for assisted screening of cervical cancer. Unlike existing approaches which are dependent on segmentation of individual cells, our approach classifies images with multiple cells. In our knowledge this is the first strictly segmentation free cervical cancer detection algorithm in the literature. This approach of segmentation free classification will lead to massive reduction of effort in collecting training and testing data. By classifying whole images, we are also able to use the extracellular information directly in the algorithm. In tune with the requirements of assisted screening, we use the intrinsic information in the neural network to provide the doctor with a map of important images and regions inside an image. While this map improves the interpretability of our results, it also speeds up the review process for the doctor. The proposed map has sub-image resolution while the training data only contains image level annotations. In our knowledge, this is the first work in classification of cervical cancer that works at image level, use extracellular information and gives results with higher spatial resolution than the training data.

## 2 Related work

During the past decade, extensive research has been devoted towards automating PAP-test. A common characteristics of these approaches have been the focus on classification of individual cells [4] [5]. This focus introduces challenges in localisation and segmentation of single cells. It also raises the challenge of integrating extracellular information to give a result at slide/image level. Hence use of these algorithms in a clinical device would necessitate accurate segmentation algorithms for delineating single cells as well as information aggregation modules to integrate the extracellular and intracellular information.

Segmentation algorithm for PAP-test images should be able to separate single cells from the background, artefacts and other cells in large images. Herlev dataset [6], which is the most popular dataset for PAP-test consists of single cells with a small amount of background. Hence the algorithms evaluated only on Herlev dataset cannot be used as a prior segmentation step for classification of cervical cells. Even discounting this fact, segmentation algorithms in literature [5, 7, 8, 9] do not meet the required level of accuracy [10].

Another approach of classifying image patches with single cells [11, 12, 10] show impressive results without the need for segmentation. Though this approach do ameliorate the need for segmentation, it needs a prior detection or localisation algorithm for localising single cells.

From the medical literature [13], it is evident that the PAP-test screening is based on the features of cells in context of extracellular information. Unfortunately the prevalent practice of classifying single cells exclude the use of extracellular information. An illustrative example for extracellular information would be the presence of inflammatory cells.

## 3 Our contributions

In this work, we propose a novel deep neural network based algorithm for classification of PAP-test images along with an assistive map that provides inter-image score for ranking images and intra-image ranking for patches in terms of their usefulness in diagnosis.

The proposed networks acts on images that contain multiple cells along with other artefacts and predicts a label for the whole image. This is in contrast with other algorithms in literature, which focused at classification of cells. In cell based approach, each cell has to be delineated and labelled. Delineation and labelling are quite time consuming, especially since the labelling need experienced pathologists. By classifying tiles instead of cells, we reduce the time and effort to collect the data. We also hypothesise that since a image consists of cells in the local context, the image based decision approach would be able to use the contextual information like the presence of neutrophils.

The proposed network provides classification label along with an assistive map for each image. The network is able to classify each image into one of the two classes, normal and abnormal. The abnormal class consists of low-grade squamous intraepithelial lesion (LSIL), high-grade squamous intraepithelial lesion (HISIL) and Squamous cell carcinoma(SCC) while normal class consists of the normal cells, all according to the Bethesda system [14]. The assistive map acts as an assistive tool for the pathologist. The map provides a score for each $30 \times 30$ pixel block in the image. The sum of scores of all the constituent blocks is the net score (assistive score) for an image. Unlike segmentation algorithms which aim to accurately segment cells, assistive map highlights regions/images that contribute to classification. We hypothesise that by presenting images in order of assistive score, a pathologist would be able to confirm the classification results by viewing minimal number of images.

**Summary of contributions**

1. Strictly segmentation free PAP-test image classification

2. Assistive map that helps pathologist focus on important tiles and regions.

3. A workflow that will reduce the workload of pathologist

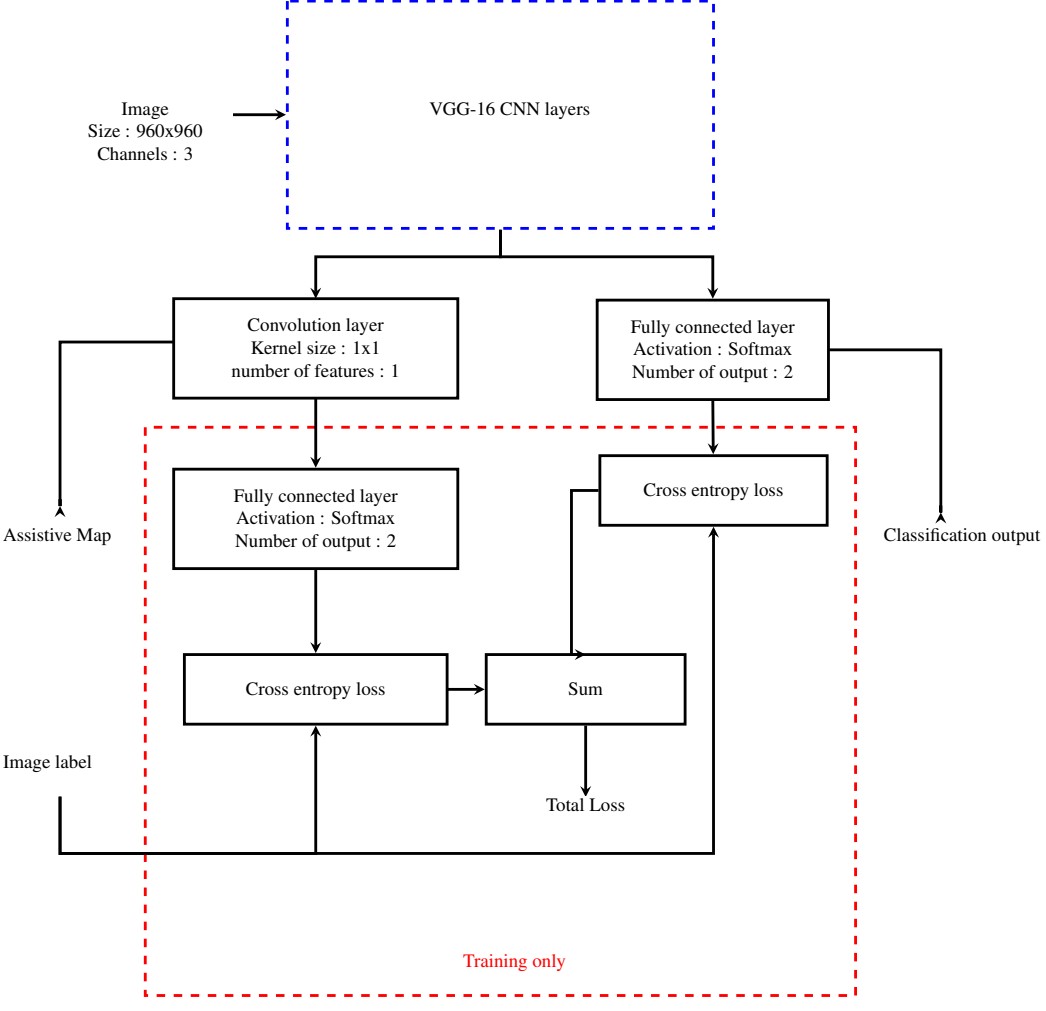

Figure 1: Network architecture

# 4 Proposed Network

The proposed network consists of a CNN feature extractor followed by a classification network in parallel to the assistive map network Figure 1. The CNN network consist of block-1 to block-5 of VGG-16 network. We use the output of block5-maxpool layer as the features from the CNN layers. For the classification we use a simple network with two neuron fully connected layer with softmax activation.

The assistive map network is designed to give a score for small blocks in the image. The standard size of a cell at 40x magnification is around $30 \times 30$ pixels. The features of the VGG CNN layer at block-5 maxpool layer has an effective receptive field of $32 \times 32$. Hence the assistive network uses these features for the map generation. The assistive network consists of a $1 \times 1$ convolution layer with just 1 feature output dimension. By using this architecture, we force the network to give a single representative value for each $32 \times 32$ pixel block. To ensure that these values are non-negative but unbounded in magnitude, we use Relu as the activation function for this layer. Though the assistive network is complete at this level, we cannot train the said network since the training data does not contain annotation for $32 \times 32$ pixel blocks. To train the assistive network with image level labels, we append a two neuron fully connected layer with softmax activation on top of the network. We use the actual tile label as the target for this layer while training. We hypothesise that since the final layer is just a softmax function, and due to the bottleneck at assistive layer, the network will train in

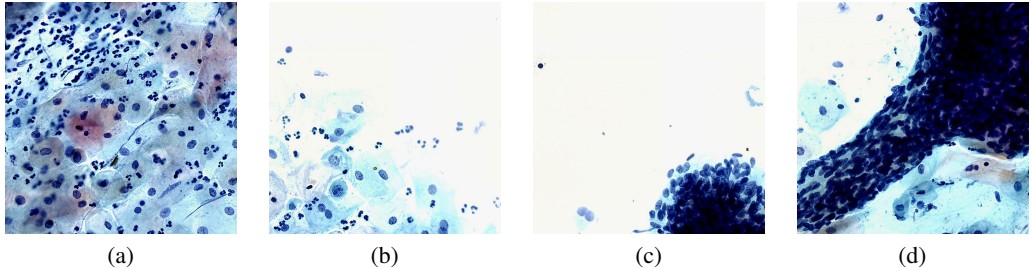

| (a) | (b) | (c) | (d) |

Figure 2: Sample images from the dataset that demonstrate variability in the distribution of cells, presence of neutrophils etc

such a way as to have the output of assistive map to be the likelihood of the region being normal or abnormal.

For training, we use categorical cross entropy loss. We compute the cross entropy loss for both the networks and sum them to get the final loss. We use the Adam optimiser[15] with a learning rate of $10^{-5}$ to train the network. Since the network was not optimised for performance, we do not report training/testing speeds or model size.

## 5 Experiments and results

The proposed network for joint classification and assistive map generation for PAP-test images is the first of its kind. The existing datasets for classification of PAP smear are not suitable for evaluating the network as they contain single cells. While the datasets for segmentation like the ISBI overlapping cervical cell segmentation challenge consists of multiple cells, the dataset does not have labels for stages of cancer and consequently classification. Hence we benchmark the proposed algorithm on a new dataset consisting of PAP smear images and the corresponding annotations.

### 5.1 Dataset

The proposed dataset is designed to evaluate algorithms that classify PAP-test images. The dataset consists 14301 images of size $961 \times 961$ pixels. These images are from conventional PAP smear slides digitised at a magnification of 40x. Some sample images from the dataset is given in Figure 2.

PAP smear usually contains large regions that are not usable for diagnosis. We reject these images from the our classification experiments. Among the leftover, we use the popular data collection strategy as exemplified by Herlev dataset. Each of the images are annotated by three experienced pathologists. We select only those images where all three pathologists agree to be part of dataset used to evaluate classification. Following this process, out of the original 14301 images 7745 images are not usable, 1220 images had disagreement between annotators, 5432 images had annotation agreed upon by annotators. Following the practice in literature, we balance the final 5432 images to contain equal number of normal and abnormal images. Hence our final dataset for classification consist of 1124 images equally distributed between normal and abnormal classes.

To evaluate the assistive network, we utilise the full set of images. The dataset consists of two class, class A and B. Class A consists of all images rejected as not usable for diagnosis while class B consists of the rest. We club the images where pathologists agree along with those on which all of them do not agree into class A. This is because disagreement between pathologists on the type of cells do not discount the usefulness of the image in diagnosis.

### 5.2 Assistive map results

#### 5.2.1 Using assistive map for sub-tile regions

The output of assistive network can be seen in Figure 3. In Figure 3 first column shows images from the dataset and the second column shows the respective images assistive maps. The third column

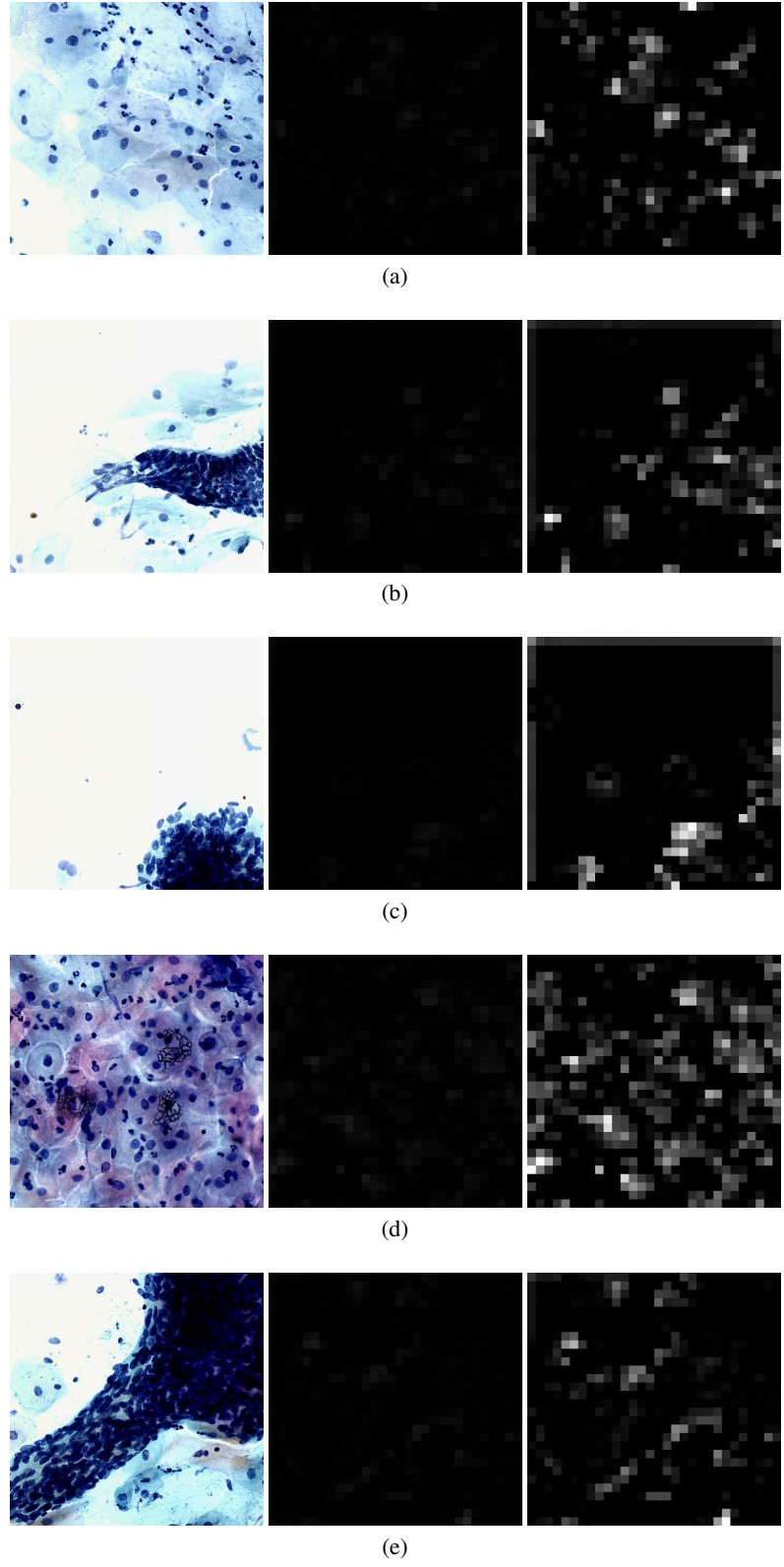

Figure 3: Images and their corresponding assistive maps. The first column contains the original images, the second row contains the respective assistive maps. The third column is the normalised version of the assistive map where the minimum value has been set to zero and maximum to 255.

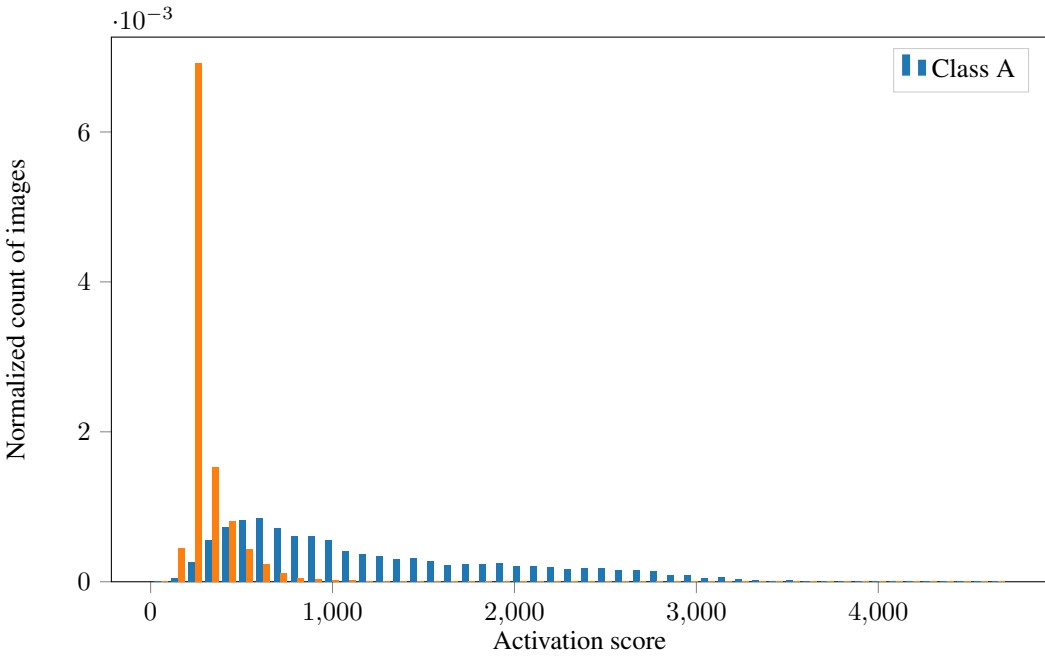

Figure 4: Normalised histogram of the number of images of each class against their assistive score. The histogram is normalised such that the area under histogram of each class sum to one. It can be clearly observed that most of the class B images have very low assistive score

shows normalised assistive map where we modified each map to have a minimum value of zero and maximum 255 for the purpose of visibility. From the Figure 3 is clear that assistive map is able to identify relevant patches in the images.

In the images 3a and 3b with lesser cell concentration towards bottom left and top left respectively, we can see that the assistive map follows the trend. This phenomena can be seen much more clearly in 3c. The image corresponding to 3c has a blob of cells at the bottom right corner. While the center of the blob is quite dense with no clear cells, the surrounding portion consists of less dense but clear cells. In the assistive map of 3c, this distribution manifests as a bright region with dark inner core and exterior. The same phenomena can be seen in 3e where the edge of the diagonal shaft is bright in the assistive map. We also show an image with uniform distribution of cells in 3d. As expected, the corresponding activation map shows a uniformly bright map.

The results in Figure 3 clearly shows that the assistive map is able to identify relevant regions inside a tile. Since the dataset annotation is only at tile level, we are unable to provide quantitative results that demonstrate the utility of assistive map for subtile regions. However an indirect quantitative result can be seen in section 5.2.2

### 5.2.2 Using assistive map for inter-tile ordering

The assistive map has the capability to identify relevant regions inside of each image. However, much more important from the perspective of assistive screening is the ability to score images on their importance to screening. By the estimate based on our observations while collecting dataset, about half of the total images are not usable for diagnosis. If we are able to use the assistive score to present images such that the pathologist is able to see most of the usable images before the unusable ones, work load of the pathologist would be reduced significantly.

The Figure 4 shows the histogram of the assistive score for each image for class A and B. The top row shows the normalised counts and bottom row shows the raw counts. From the normalised histogram it is clear that most of the class B images have low assistive score in comparison to the class A images. Hence the proposed workflow can potentially work. The Figure 5 shows a further validation of this point. Figure 5 shows the percent of images from each class among the images pathologist has seen

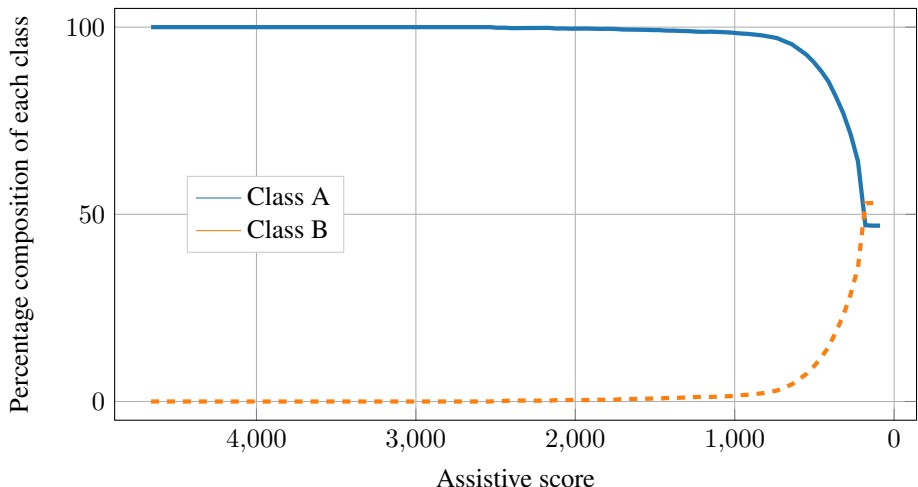

Figure 5: Percentage of total images from each class seen by pathologist when presented in the decreasing order of assistive score. Note that the percentage of class A images are high at the start

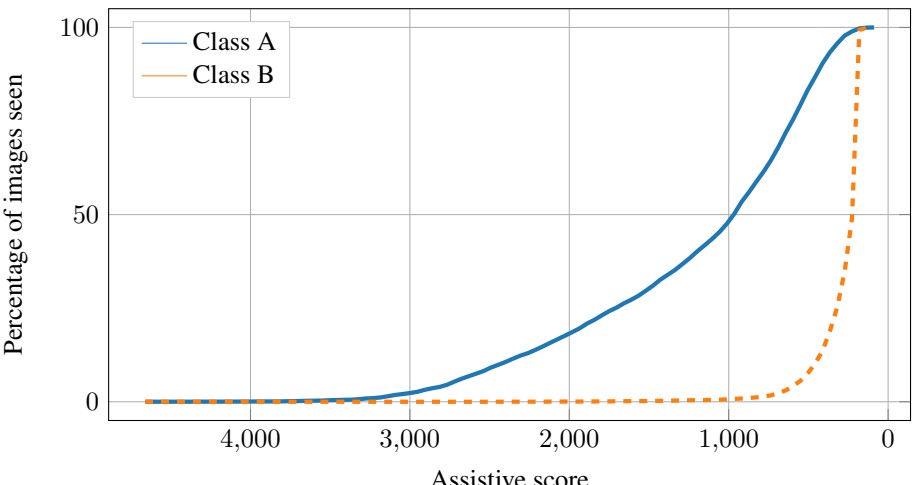

Figure 6: Percentage of total images of each class seen by pathologist when presented in the decreasing order of assistive score

when presented in the inverse order of assistive score. The figure clearly shows that most of the images that pathologist has seen at the start are useful images.

To further quantify the use of assistive score, we can look at how much percentage of the total images in each class the pathologist would have seen at any point in time. This is given in Figure 6. This figure reinforces our approach. It can be clearly seen that the pathologist is able to review more good images than bad images initially. In fact according to the figure, by the time the pathologist reviews 90% of good images, he would come across only around 10% of bad images.

## 5.3 Classification results

The results of tile classification can be seen in Table 1. From the table we can see that we are able to achieve a relatively high accuracy of 83% on the data considering the fact that PAP-smear test is highly subjective. Since there are no other works in literature to compare, we do not analyse the results in detail.

Table 1: Classification performance

| Accuracy |
|----------|
| 83.01% |

## 6   Conclusion

In this work we propose a deep neural network for assistive screening of cervical cancer. The proposed network assists screening by providing a classification at image level along with an assistive map that helps in evaluating the usefulness of a tile. We propose using the output of classification network for initial results and the assistive network to prioritise review of the these results, thereby reducing the work load on pathologist. Our experiments and the subsequent analysis clearly shows that the proposed network is able to achieve the target of reducing pathologist workload.

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
