# OpenReview forum: "Deep neural network based assistive screening for cervical cancer"
_MIDL.amsterdam/2018/Conference — Submitted to MIDL 2018_

### Review · AnonReviewer2 · 2018-05-05
**a standard approach that learns a deep network to classify cervical cancer for assistive screening**

**Rating:** 2
**Confidence:** 3

**Review:**

The paper presents a neural network approach for classifying cervical cancer using PAP smear image. The network is based on VGG. The assistive map generation is essentially the global average pooling.

The paper lacks novelty in methodology. In experiment, it lacks comparison with other state-of-the-art networks.

From Figure 3, the assistive maps do not necessarily correspond to the images.

**Special Issue:**

No

---

### Review · AnonReviewer3 · 2018-05-08
**workflow, but not algorithm**

**Rating:** 1
**Confidence:** 2

**Review:**

A workflow is presented that helps speeding up histopathology classification by experts. Important building block is a saliency map that visualizes relevant areas using a network trained on global labels only.

Pro: I would share the authors view that leveraging time and expertise of domain experts will be of larger importance that aiming at fully automated processing algorithm (with unknown behaviour in case of failure). As such I liked the paper, application, and contribution brought forward.

Con: I would not see how the present work advances the current state of research in algorithmic development, but rather see this as a purely application dominated contribution. As such, I would rather see it as a contribution to a application/pathology related venue.

**Special Issue:**

No

---

### Review · AnonReviewer1 · 2018-05-17

**Rating:** 1
**Confidence:** 3

**Review:**



**Special Issue:**

No

---

### Decision · Program_Chairs · 2018-05-15
**Paper49 Acceptance Decision**

Reject